# Relationships between Causal Factors Affecting Future Carbon Dioxide Output from Thailand's Transportation Sector under the Government's Sustainability Policy: Expanding the SEM-VECM Model

**Pruethsan Sutthichaimethee *** and Danupon Ariyasajjakorn

Faculty of Economics, Chulalongkorn University, Wang Mai, Khet Pathum Wan, Bangkok 10330, Thailand; danupon.a@chula.ac.th

\* Correspondence: pruethsan.sut@gmail.com; Tel.: +66-639-645-195

**Abstract:** This research aims to analyze the relationships between causal factors likely to affect future $CO_2$ emissions from the Thai transportation sector by developing the Structural Equation Modeling-Vector Autoregressive Error Correction Mechanism Model (SEM-VECM Model). This model was created to fill information gaps of older models. In addition, the model provides the unique feature of viable model application for different sectors in various contexts. The model revealed all exogenous variables that have direct and indirect influences over changes in $CO_2$ emissions. The variables show a direct effect at a confidence interval of 99%, including per capita GDP ($\Delta\ln(GDP)_{t-1}$), labor growth ($\Delta\ln(L)_{t-1}$), urbanization rate factor ($\Delta\ln(URT)_{t-1}$), industrial structure ($\Delta\ln(IS)_{t-1}$), energy consumption ($\Delta\ln(EC)_{t-1}$), foreign direct investment ($\Delta\ln(FDI)_{t-1}$), oil price ($\Delta\ln(OP)_{t-1}$), and net exports ($\Delta\ln(X-E)_{t-1}$). In addition, it was found that every variable in the SEM-VECM model has an indirect effect on changes in $CO_2$ emissions at a confidence interval of 99%. The SEM-VECM model has the ability to adjust to the equilibrium equivalent to 39%. However, it also helps to identify the degree of direct effect that each causal factor has on the others. Specifically, labor growth ($\Delta\ln(L)_{t-1}$) had a direct effect on per capita GDP ($\Delta\ln(GDP)_{t-1}$) and energy consumption ($\Delta\ln(EC)_{t-1}$) at a confidence interval of 99%, while urbanization rate ($\Delta\ln(URT)_{t-1}$) had a direct effect on per capita GDP ($\Delta\ln(GDP)_{t-1}$), labor growth ($\Delta\ln(L)_{t-1}$), and net exports ($\Delta\ln(X-E)_{t-1}$) at a confidence interval of 99%. Furthermore, industrial structure ($\Delta\ln(IS)_{t-1}$) had a direct effect on per capita GDP ($\Delta\ln(GDP)_{t-1}$) at a confidence interval of 99%, whereas energy consumption ($\Delta\ln(EC)_{t-1}$) had a direct effect on per capita GDP ($\Delta\ln(GDP)_{t-1}$) at a confidence interval of 99%. Foreign direct investment ($\Delta\ln(FDI)_{t-1}$) had a direct effect on per capita GDP ($\Delta\ln(GDP)_{t-1}$) at a confidence interval of 99%, while oil price ($\Delta\ln(OP)_{t-1}$) had a direct effect on industrial structure ($\Delta\ln(IS)_{t-1}$), energy consumption ($\Delta\ln(EC)_{t-1}$), and net exports ($\Delta\ln(X-E)_{t-1}$) at a confidence interval of 99%. Lastly, net exports ($\Delta\ln(X-E)_{t-1}$) had a direct effect on per capita GDP ($\Delta\ln(GDP)_{t-1}$) at a confidence interval of 99%. The model eliminates the problem of heteroskedasticity, multicollinearity, and autocorrelation. In addition, it was found that the model is white noise. When the SEM-VECM Model was used for 30-year forecasting (2018–2047), it projected that $CO_2$ emissions would increase steadily by 67.04% (2047/2018) or 123.90 Mt $CO_2$ Eq. by 2047. The performance of the SEM-VECM Model was assessed and produced a mean absolute percentage error (MAPE) of 1.21% and root mean square error (RMSE) of 1.02%. When comparing the performance value with the values of other, older models, the SEM-VECM Model was found to be more effective and useful for future research and policy planning for Thailand's sustainability goals.

**Keywords:** $CO_2$ emissions; SEM-VECM model; long-term relationship; economic growth; policy modeling

## 1. Introduction

In recent years, Thailand has been experiencing steady economic growth as evidenced by the Gross Domestic Product (GDP) growth from 1997 to present [1]. Much of the revenue growth contributing to Thailand's economy can be attributed to government policies promoting foreign investments in Thailand, supporting exports in all sectors, supporting the tourism industry, reducing reliance on imports, encouraging domestic investment in SMEs, and raising the minimum wage [2]. Such policies have helped Thailand to develop its economy while boosting living standards at the same time (2000–present) [1,2]. However, this has also led to an increase in energy consumption (1997–present), as well as a steady rise in $CO_2$ emissions.

Considering all sectors, transportation is ranked amongst the highest in energy consumption [3] and $CO_2$ emissions. As a result, it produced 75 Mt $CO_2$ Eq. of greenhouse gasses in 2017 with a growth rate of 17.1% compared to 2016 [4]. The sector was found to produce up to 91.2 percent of all greenhouse gases in 2017, a percentage that has increased steadily since 1997 [3–5].

As a result of the increase in the transportation sector's greenhouse gas emissions, a new $CO_2$ forecasting model that is accurate, efficient, and effective is required to guide the formulation of policy and planning. To this end, the SEM-VECM Model was developed from existing models to be made available for national planning purposes, as well as for future application in other fields. The researchers reviewed a number of relevant studies focusing on two important components, namely relationship factors and forecasting models, to serve as guidance in the research process. A streamlined review of studies was conducted for the relational investigation. Hu et al. [6] investigated the relationship between energy consumption and economic growth in industrial sectors in China by using first-and-second-generation panel unit root tests, panel co-integration tests, and a system generalized moment method. The study found that, in the short term, there was a unidirectional causal relationship between economic growth and energy consumption. Furthermore, in the long term, a unidirectional causality was found between energy consumption and economic growth. Zhao et al. [7] produced a comparative study investigating the equilibrium relationships and causal relationships between economic growth, electricity consumption, labor force, and capital input in northern China by applying a panel data analysis method based on the Cobb-Douglas production function. Their findings showed all variables to be long-term co-integrated. In addition, bidirectional causal relationships were found between electricity consumption and economic growth in six of these provinces, excluding the Hebei province. The same study also showed a bidirectional relationships between capital input and economic growth, as well as between labor force and economic growth, except for in the Beijing and Hebei province. Armeanu et al. [8] attempted to explore the influence and causal relation between renewable energy and sustainable economic growth in the 28 countries of the European Union (EU) during 2003–2014 by using a multivariate panel data. In their study, they noted that biomass energy had the highest influence on economic growth, while there was an indication of a unidirectional causal relationship in the short and long term between sustainable economic growth and renewable energies. Bandalos [9] examined the preciseness and utility of overall error and error estimators in the structural equation models by using a method of Monte Carlo. In the study, it has shown that the rescaled non-centrality parameter and EFO produced a highly precise estimate of the approximation error and overall error amount. Gómez et al. [10] investigated the linear and nonlinear causality relationship between energy consumption and economic growth in Mexico from 1965 to 2014 by employing unit root with structural breaks, co-integration analysis, and linear and nonlinear causality tests. They concluded that there were long-term linkages among production, capital, labor, and energy. In terms of linear causal links, they extended from total and disaggregated energy consumption to economic growth. Nonlinear causality went from energy consumption, transportation, capital, and labor to output. This result affirmed the importance of input factors in economic activity, and that energy conservation policies would have an impact on economic growth in Mexico.

Arango-Miranda et al. [11] produced a comparative empirical study to explore the links between carbon dioxide emissions, energy consumption and economic growth in certain developed and

developing countries. The study employed the hypothesis deigned for the Environmental Kuznets Curve (EKC). The study's findings did not support the hypothesis, yet it claimed that the exergy intensity is useful for future research. Chang [12] initiated an investigation of correlations between carbon dioxide emissions, energy consumption and economic growth in China by adapting a multivariate causality test. He claimed that a long-term carbon dioxide reduction policy would negatively affect the economy due to a closed-form relationship among the variables, and additionally, that economic growth would raise energy consumption and $CO_2$ emissions, worsening global climate change. In six sub-Saharan African nations, Kivyiro and Arminen [13] analyzed the casual relationships between carbon dioxide emissions, energy consumption, economic growth, and foreign direct investment by implementing an autoregressive distributed lag model and co-integration. The results indicated that all variables are co-integrated in the long term among all countries, and in certain countries, foreign direct investment (FDI) was found to have a greater potential to raise $CO_2$ emissions. In general, unidirectional Granger causality links were observed regarding the relation between the other variables and $CO_2$ emissions. In other studies, the same areas of relational investigation have been explored. Wesseh and Zoumara [14] examined the causal independence between energy consumption and economic growth in Liberia by investigating evidence of a non-parametric bootstrapped causality test. The study contributed to the existence of distinct bidirectional Granger causality between the variables. Additionally, it examined how employment in Liberia influences economic growth, while suggesting the appropriateness of the bootstrap technique. Yoo and Ku [15] investigated the causal relationship between nuclear energy consumption and economic growth in six countries, namely Pakistan, Switzerland, Argentina, Korea, France, and Germany, by using time-series techniques of unit roots, co-integration, and Granger-causality. In this study, it was found that the relationships between the variables among the countries were not uniform. Two types of relationships were observed, presenting both bidirectional causality between nuclear energy consumption and economic growth in Switzerland, and unidirectional causality between economic growth and nuclear energy in the case of France and Pakistan. Moreover, the same causality was found in Korea leading from nuclear energy to economic growth. While two other countries, Argentina and Germany, did not display any of those relationships. Chang et al. [16] examined G6 countries, analyzing the causal relationship between nuclear energy consumption and economic growth in those countries by optimizing the panel Granger causality tests. The findings of the study showed a unidirectional causality from economic growth to nuclear energy consumption across the countries, however, the UK was found to have a bidirectional causality from nuclear energy consumption to economic growth.

In addition to the above studies, Nasreen and Anwar [17] explored the causal relationships between trade openness, economic growth and energy consumption in Asian countries by applying panel co-integration and causality approaches. The study concluded that all variables are co-integrated and bidirectional causality is present between them. It also concluded that trade openness has a positive impact on energy consumption. With regards to China, Zhixin and Xin [18] studied the causal relationships between energy consumption and economic growth in the Shandong province by utilizing statistical data from 1980 to 2008 with an adaptation of unit root, co-integration, and the Granger causality test. The study indicated a long-term relationship and two-way causality between the two factors, while also showing a positive relationship between them within the province and that economic growth is strongly dependent on energy consumption. Yu et al. [19] focused on studying and projecting urban energy consumption and $CO_2$ emissions in Beijing from 2005–2011 and 2012–2030 by developing the Long-range Energy Alternatives Planning System (LEAP)-BJ model. The results of the study showed how incremental changes in energy consumption led to fluctuations in total $CO_2$ emissions during 2005–2011. It was estimated that Beijing would reduce total energy consumption by 21.36% and $CO_2$ emissions by 35.37% from 2012 and 2030 if the proposed policies were implemented in full under the POL scenario. Mudarissov and Lee [20] examined the casual relationship between energy consumption and economic growth in Kazakhstan by adapting various methods, including Granger causality, the Vector Error Correction Model, an augmented Dickey–Fuller and Phillips–Perron

unit root tests, and a co-integration test. Their findings indicate long-term unidirectional causalities leading from energy consumption (EC) to economic growth, yet also reflect short-term unidirectional causalities leading from economic growth to energy consumption. This indicates the significance of national energy production in boosting the economic growth.

A number of studies have employed various models for forecasting purposes. In Shandong, China, Li and Li [21] forecasted energy consumption using the autoregressive integrated moving average (ARIMA) Model, Gray Model (GM (1,1)), and the ARIMA-GM Model. The study concluded that energy consumption during the forecasted years would increase at an average annual rate of 3.9 per cent, and that by 2020, the province's energy demand would climb about 20 per cent compared to that of 2015. Ozturk and Ozturk [22] forecasted Turkey's energy consumption for a 25 year period using the ARIMA Model. By using this model, they were able to demonstrate that Turkey's energy consumption could be expected to rise continuously until 2040, with the consumption of coal, oil, natural gas, renewable energy, and total energy growing at an annual average percentage of 4.87, 3.92, 4.39, 1.64, and 4.20, respectively. In recent years, numerous countries have instituted policies to increase industrial production, generally requiring higher energy consumption, to boost economic growth. Hence, forecasting increases in energy consumption in a more accurate way would help facilitate governments in determining the most appropriate policies to achieve their goals. Sun et al. [23] sought to improve the accuracy of fossil fuel energy consumption predictions with regards to China's power generation. They established a novel hybrid quantum harmony search algorithm-based LSSVM (QHSA-LSSVM) Model. Yuan et al. [24] presented a comparative study on primary energy consumption forecasting in China from 2014 until 2020 with an application of the ARIMA Model and GM (1,1) Model. Their study concluded that the growth rate of the primary energy use in China during the forecasted years would increase, yet at a slower rate than the first decade of the new era. Sen et al. [25] produced a study predicting energy consumption and Greenhouse Gas (GHG) emissions for a pig iron manufacturing organization in India. Their study applied the ARIMA Model and the validity of the model indicated that it was the best-fitted model for estimating the energy consumption. Fan et al. [26] conducted a study forecasting natural gas demand in China from 2011 to 2017 by implementing a combined forecasting model called the Grey Model and Self-Adapting Intelligent Grey Model with Genetic Algorithm and Annual Share Changes (GM-S-SIGM-GA). The study indicated that the combined model outperforms any other single forecasting model, while reflecting the following values; mean absolute percentage error (MAPE; 4.48%), root mean square error (RMSE; 11.59), and mean absolute error (MAE; 8.41), respectively.

In order to determine other associations for energy consumption prediction, Barak and Sadegh [27] proposed an ensemble ARIMA–adaptive neuro-fuzzy inference system (ANFIS) hybrid algorithm for predicting Iran's energy consumption. The results indicated that the models could be used to enhance the accuracy of a single ARIMA or ANFIS in the prediction of energy consumption. Okumus and Dinler [28] attempted to combine the ANFIS with an artificial neural network (ANN) for wind-speed forecasting and wind power generation. The prediction results provided a MAPE of 2.2598%, 3.3530%, and 3.8589% at three different locations for daily average wind speeds. Zeng et al. [29] constructed a forecasting model called the Homologous Grey Prediction Model which was used to forecast energy consumption in China's manufacturing sector for the years 2018 to 2024. The final prediction result reflected a significant downward trend in the total energy consumption during the forecasted years. Dai et al. [30] proposed a new forecasting model embracing various approaches named the Ensemble Empirical Mode Decomposition and Least Squares Support Vector Machine Optimized by Improved Shuffled Frog Leaping Algorithm (EEMD-ISFLA-LSSVM) to accurately predict energy consumption in China from 2018 to 2022. Their study showed that there would be significant growth potential in China's energy consumption during the estimated duration. Additionally, Liu et al. [31] proposed a forecasting model by optimizing the techniques of Multi-Variable Linear Regression (MLR) and Support Vector Regression (SVR) along with a Gated Recurrent Unit (GRU) Artificial Neural Network in order to predict the Chinese primary energy consumption from 2015 to 2012. Their prediction was a

drop in energy consumption from 2954.04 Mtoe to 5618.67 Mtoe by 2021. Ma et al. [32] utilized linear (Metabolic Grey Model), nonlinear (Non-linear Grey Model), and combined models (Metabolic Grey Model-Autoregressive Integrated Moving Average Model) to predict the coal consumption in South Africa from 2017 to 2030. The final result indicated a downward trend in future coal consumption during the forecasted period, decreasing by 1.9 per cent per year. Ma et al. [33] attempted to forecast renewable energy consumption by applying a machine learning forecasting algorithm devoid of massive independent variables and assumptions. They concluded that the said technique produces an enormous improvement of up to ~138.26-fold on REC-BMs and ~24.67-fold on HE-EC. At the same time, they estimated that the technique would be able to save the USA ~2692.62 PJ petajoules (PJ) on HE-EC and ~9695.09 PJ on REC-BMs for the eight-year forecast period.

By reviewing the relevant literature, we were able to determine that a number of studies have been conducted on the same research area as ours, yet using different approaches. Furthermore, we discovered that there were no studies modeling $CO_2$ emissions based on the SEM-VECM Model. Thus, we developed a model to use in sustainable development planning for Thailand in order to maximize useful data outcomes. We used the time series data from 1990 to 2017 in Thailand to do a forecast of $CO_2$ emissions for the next 30 years (2018–2047). The projected results are to be used in the policy planning of Thailand based on the next 30 years' (2018–2047) sustainable development strategy. The selection of independent variables was based on the framework of the above management strategy. Hence, this research will be deemed useful and beneficial to national management and future applications. The research process was as follows:

1. Identify a variable framework according to Structure Equation Modeling [34], where exogenous variables and endogenous variables are extracted to be latent variables and observed variables.
2. Analyze the long-term relationships of the causal factors based on the theory of Augmented Dickey-Fuller [35,36] optimized with the concept of Johansen and Juselius [37,38].
3. Choose variables that have a co-integration at the same level to construct a SEM-VECM Model where the relationship of causal factors is both in the short and long term, indicating the direct effect, indirect effect, and total effect of the relationship.
4. Examine the developed model regarding its heteroscedasticity, multicollinearity, and autocorrelation.
5. Compare the effectiveness of the SEM-VECM Model with other existing models, including Multiple Linear Regression, Gray Model (GM (1,1)), GM-ARIMA Model, Artificial Neural Natural Model (ANN), back propagation neural network (BP Model), and ARIMA Model, through the performance measures of MAPE and RMSE.
6. Analyze the relationship and direction parameter estimates of the SEM-VECM Model.
7. Forecast $CO_2$ emissions for the next 30 years (2018–2047) using the SEM-VECM Model. The flowchart of the SEM-VECM Model is shown in Figure 1 below.

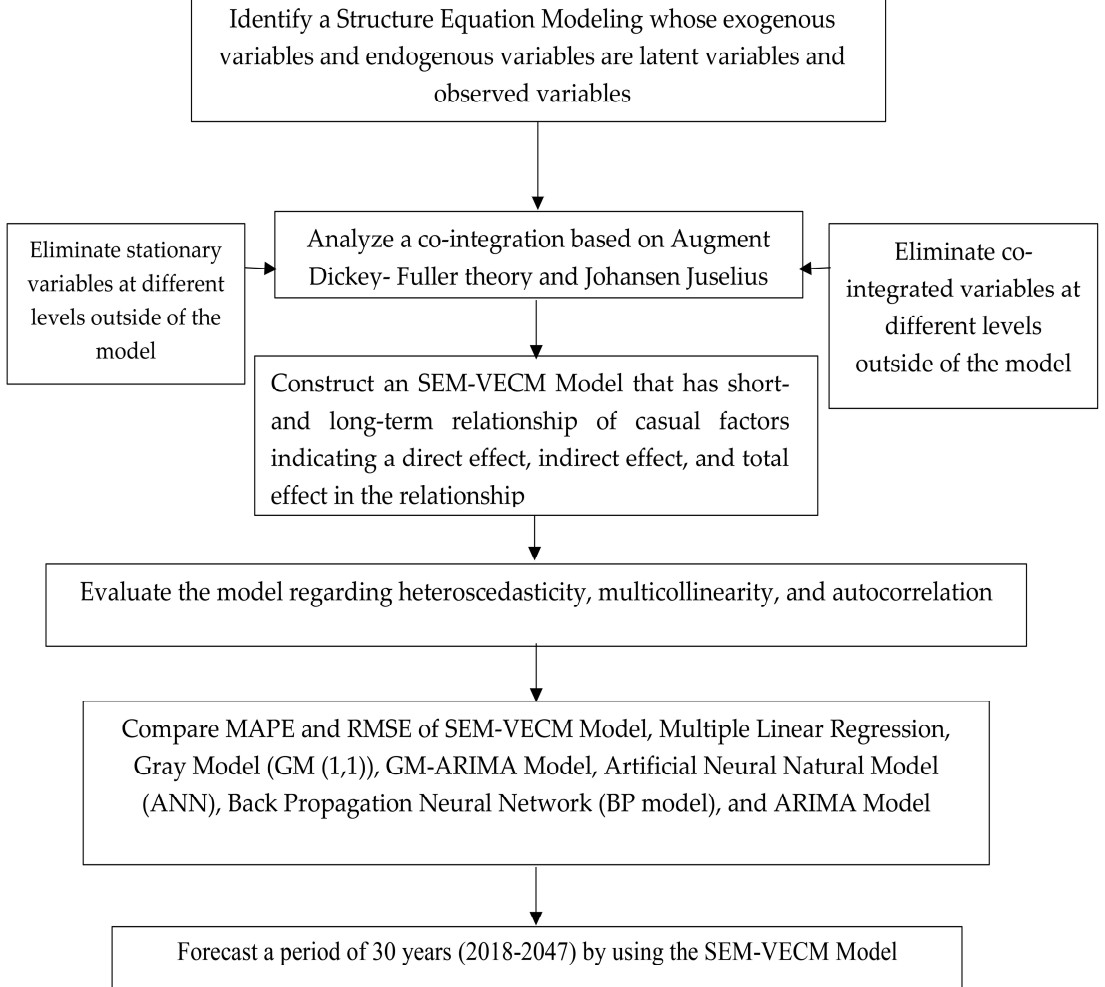

**Figure 1.** The flowchart of the Structural Equation Modeling-Vector Autoregressive Error Correction Mechanism (SEM-VECM) model.

## 2. The Forecasting Model

### 2.1. Structure Estimation Modeling-Vector Error Correction Mechanism Model (SEM-VECM Model)

The SEM-VECM Model is adapted from the theory of Structure Estimation Modeling (SEM) [34] with an optimization of the Vector Error Correction Mechanism Model (VECM) [39]. The model is illustrated in Figure 2.

When we align all relationships in Figure 2, it can be seen that there are independent variable ($\xi$), dependent variable ($\eta$), observed independent variable ($X$), observed dependent variable ($Y$), variance of estimated dependent variable ($\zeta$), variance of observed variable $X$ ($\delta$), variance of observed variable $Y$ ($\varepsilon$), correlation coefficient between independent variables ($\phi$), correlation coefficient between independent variable and dependent variable ($\gamma$), and correlation coefficients between dependent variables ($\beta$).

Where

$\xi$ = independent variable (exogenous construct)

$\eta$ = dependent variable (endogenous construct)

$X$ = observed variable of $\xi$

$Y$ = observed variable of $\eta$

$\lambda_X$ = correlation coefficient between independent variable and observed variable

$\lambda_Y$ = correlation coefficient between dependent variable and observed variable

$\delta$ = variance of estimated variable $X$

$\varepsilon$ = variance of estimated variable $Y$

$\phi$ = correlation coefficient between independent variables

$\beta$ = correlation coefficients between dependent variables

$\gamma$ = correlation coefficient between independent variable and dependent variable

$\zeta$ = variance of estimated dependent variable

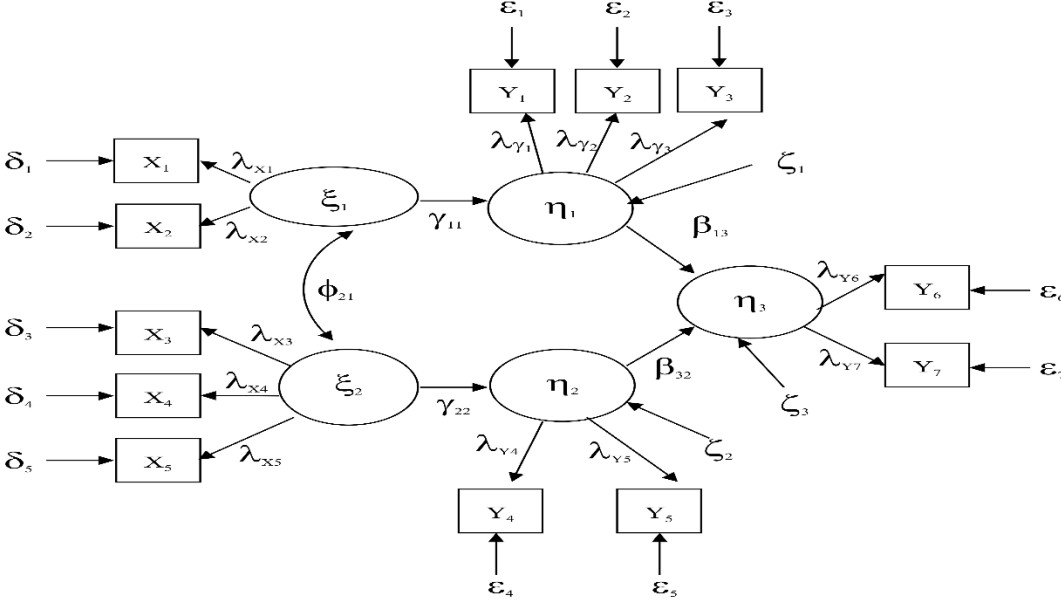

**Figure 2.** Relationships of all variables in the model.

However, the SEM model incorporates certain variables that fall under causal factors, yet are stationary as conceptualized by Dickey and Fuller [35]. When all causal factors are found to be stationary, an analysis of co-integration test based on Johansen and Juselius [36] has to be put in place. Then, the outcome was used to form the SEM-VECM model to estimate the size of parameter, as well as construct a forecasting model as shown in the following.

In this section, there are certain areas we must explore before analyzing the data. Here, the stochastic or deterministic data has the potential to make the model spurious. The spurious model is found when the estimated value is well-correlated, yet it does not reflect a true magnitude of influence. This is due to the data that were used in the modeling being non-stationary. Dickey and Fuller [35] introduced a theory to tackle the issue by undertaking a unit root test called the Augmented Dickey-Fuller test for non-stationary data, the results of which can be compared with MacKinnon Critical Values [36] as illustrated below:

$$X_t = \rho X_{t-1} + \varepsilon_t \tag{1}$$

We take Equation (1) to perform the unit root test. If the output gives $|\rho| < 1$, the data is stationary. In contrast, if $\rho = 1$, $X_t$ is non-stationary. From the same equation, if the random walk with drift joins in the equation, the output will be as follows:

$$\Delta X_t = \alpha + \rho X_{t-1} + \varepsilon_t \tag{2}$$

In Equation (2), if the random walk with drift and linear time trend are considered in the equation, the above equation will be as follows:

$$\Delta X_t = \alpha + \beta t + \delta X_{t-1} + \varepsilon_t \tag{3}$$

where $t$ = time, and $\rho = (1 + \delta)$. If $\delta$ is negative, $X_t$ exists at the integration of order zero.

Hence, Equations (1)–(3) were put forth for the co-integration test of at least two variables. Here, we used $X_t$ and $Y_t$. If $X_t$ and $Y_t$ are integrated of order d, it can be written as $I(d)$, and it must have a linear combination as $\alpha X_t + \beta Y_t$ is integrated of order $(d\text{–}b)$, given that $d > b > 0$.

Therefore, implementing the co-integration test based on Augment Dickey-Fuller can reflect a serial correlation as shown below:

$$\Delta \hat{e}_t = \gamma \hat{e}_{t-1} + \sum_{i=1}^{p} \phi_i \Delta \hat{e}_{t-1} + v_t \tag{4}$$

For Equation (4), if $-2 < \gamma < 0$, the residual is stationary, and the above equation has a co-integration. Since there is the existence of co-integration, we had to take the above equation for an Error correction mechanism analysis as demonstrated below:

$$\Delta Y_t = \phi_1 + \phi_2 \hat{e}_{t-1} + \phi_3 \Delta X_t + \sum_{h=1}^{p} \phi_{4h} \Delta X_{t-h} + \sum_{i}^{p} \phi_{5i} \Delta Y_{t-i} + v_t \tag{5}$$

where $\hat{e}_t$ is residual of the co-integrating regression equation and $\phi_2$ is a discrepancy between the actual value of $Y_t$ and the long term value in the period that has been eliminated.

The SEM-VECM Model includes short-and long-term relationships indicating an equilibrum of the VECM Model in the matrix as shown below:

$$\Delta X_t = \Pi X_{t-1} + \Gamma_1 \Delta X_{t-1} + \Gamma_2 \Delta X_{t-2} + \ldots + \Gamma_{p-1} \Delta X_{t-(p-1)} + u_t \tag{6}$$

where

$\Pi = -(I - A_1 - A_2 - \ldots - A_p)$ as $n \times n$ matrix
$\Gamma_1 = (A_2 + A_3 + A_4 + \ldots + A_p)$ as $n \times n$ matrix
$\Gamma_2 = (A_3 + A_4 + \ldots + A_p)$ as $n \times n$ matrix
$\Gamma_{p-1} = -(A_p)$ as $n \times n$ matrix

Equation (6) can be drawn in the relation form between dependent variable and independent variabls as follows:

$$\Delta X_t = \alpha \beta' X_{t-1} + \Gamma_1 \Delta X_{t-1} + \Gamma_2 \Delta X_{t-2} + \ldots + \Gamma_{p-1} \Delta X_{t-(p-1)} + u_t \tag{7}$$

where $\beta' X_{t-1}$ is a $r \times 1$ vector at I(0) or written as $\beta' X_{t-1} \sim I(0)$, row 1, row 2, and row $r$ of $\beta' X_{t-1}$ shows a long-term equilibrium relationship in form 1, form 2, $\ldots$, form $r$ of time series $X_{1t}, X_{2t}, \ldots, \Delta X_{nt}$, respectively.

The number of the vector in the long-term equilibrium relationship is equal to $r$, where $r < n$. This means such a number has to be lower than the time series in the model (VECM), simplifying that the mentioned number should begin from having no vector in a long-term equilibrium relationship, $r = 0$, to the vector of $n - 1$ as $(r = n - 1)$ or written as $r = 0, 1, 2, \ldots, n - 1$.

In the case where time series $X_{1t}, X_{2t}, \ldots, \Delta X_{nt}$ have no long-term equilibrium relationship, $r = 0$, the coefficient matrix $\Pi$ becomes zero or written as $\Pi = 0$ where 0 is matrix zero. In this case, we shall use the analysis model of VAR(p) in the first difference like below:

$$\Delta X_t = A_1 \Delta X_{t-1} + A_1 \Delta X_{t-2} + \ldots + A_p \Delta X_{t-p} + u_t \tag{8}$$

In the case where time series $X_{1t}, X_{2t}, \ldots, \Delta X_{nt}$ have a long-term equilibrium relationship, the number of such relationship can be $1, \ldots, n - 1$ forms (or $r = 1, \ldots, n - 1$). Here, it can be written as $\Pi = \alpha \beta'$, where $\beta$ is $n \times r$ matrix of such a relationship.

In the case where the division of matrix $\Pi$ is a multiplied outcome of matrix $\alpha$ and $\beta'$, this can result in another $n \times r$ matrix, which can be written as $\alpha^*$ and $\beta^*$; later, such a matrix resulted in

$\Pi = \alpha^* \beta^*$. Hence, we can use $\beta^*$ to show a long-term equilibrium relationship as $\beta$. This can be proven by using any $n \times n$ non-singular matrix. This kind of matrix is denoted as $Q$ to multiply with matrix $\alpha$ and $\beta$, while leaving the coefficient matrix $\Pi$ as it is. This can be demonstrated as follows:

$$\Pi = \alpha^* \beta^{*\prime} \tag{9}$$

where $\alpha = \alpha Q'$ and $\beta^* = \beta Q^{-1}$.

Consider the SEM-VECM model as follows:

$$\Delta X_t = \alpha \widetilde{\beta}' \widetilde{X}_{t-1} + \Gamma_1 \Delta X_{t-1} + \Gamma_2 \Delta X_{t-2} + \ldots + \Gamma_{p-1} \Delta X_{t-(p-1)} + \phi D_t + u_t \tag{10}$$

where $\widetilde{\beta}' = \begin{bmatrix} \beta' & \beta_0 & \beta_1 \end{bmatrix}$ is the $r \times (n+2)$ matrix, $\beta$ is the $n \times r$ matrix, $\beta_0$ and $\beta_1$ are the $r \times 1$ vector, $\widetilde{X}_{t-1} = \begin{bmatrix} X_{t-1} & 1 & t \end{bmatrix}'$ is the $(n+2) \times 1$ vector, $\alpha$ is the $n \times r$ matrix, and rank $(\alpha)$ = rank $(\widetilde{\beta}) = r$. Additionally, $D_t$ is the matrix indicating a deterministic component.

The estimation of the parameter of the long-term co-integrating vector $\widetilde{\beta}$ can be achieved with the application of maximum likelihood by assuming vector $u_t \approx$ Normal $(0, \sum)$ 0 is zero, and $\sum$ is the variant matrix of $u_t$. Johansen [37] proved that the estimation of vector $\widetilde{\beta}_{n \times r}$ with this method would result in an eigenvector in accordance with the eigenvalue from the minimum to maximum value. This is achieved using the equation below:

$$\left| \lambda S_{11} - S_{10} S_{00}^{-1} S_{01} \right| = 0 \tag{11}$$

$$S_{ij} = \frac{1}{T} R_{it} R_{jt}', i = 0,1 \text{ and } j = 0,1 \tag{12}$$

where $T$ is the number of data used in the SEM-VECM Model. $R_{0t}$ is the $n \times T$ matrix of the residual retrieved from a regression equation with a variable of $\Delta X_t$, and the independent variable is $\Delta X_{t-1}$, $\Delta X_{t-2}, \ldots, \Delta X_{t-p+1}, D_t$. $R_{1t}$ is the $(n+2) \times T$ matrix of the residual retrieved from a regression equation with a variable of $\widetilde{X}_{t-1}$, and the independent variable is $\Delta X_{t-1}, \Delta X_{t-2}, \ldots, \Delta X_{t-p+1}, D_t$.

If $\hat{\lambda}_i (i = 1, 2, \ldots, n)^{11}$ is the eigenvalue computed from Equation (11) where $1 > \hat{\lambda}_1 > \hat{\lambda}_2 > \ldots > \hat{\lambda}_n \geq 0$, let the eigenvector consistent with the eigenvalue $\hat{\lambda}_1, \hat{\lambda}_2, \ldots, \hat{\lambda}_n$ be written as $\hat{V} = \begin{bmatrix} \hat{v}_1 & \hat{v}_2 & \ldots & \hat{v}_n \end{bmatrix}_{(n+2) \times (n+2)}$. Therefore, we can obtain the estimator of the co-integrating vector as follows:

$$\hat{V} = \begin{bmatrix} \hat{v}_1 & \hat{v}_2 & \ldots & \hat{v}_r \end{bmatrix}_{\times (n+2) \times r} \tag{13}$$

Thus, according to the concept of Johansen [37], when all variables are analyzed and found to be co-integrated, more values can be added into the error correction mechanism by inserting $ECM_{t-1}$ into a model. $ECM_{t-1}$ was incorporated into the SEM-VECM model by estimating a parameter in the same way as the other variables do. The estimated parameter will then reflect the magnitude and the adjustment ability for the equilibrium.

Therefore, the SEM-VECM Model is a model that reflects the relationship between exogenous variables and endogenous variables in the pursuit of causal relationship both directly and indirectly. Additionally, this relationship happens in both the short and long term. When the model is used for forecasting, it will be high in efficiency in outcome and viable for future application in different contexts. As for its designation, the model is most suitable for both short- and long-term prediction. Therefore, this research applied the SEM-VECM model to a 30-year forecast (2018–2047), while the model can be explained below.

Commonly, there are two popular patterns of forming primary and secondary assumptions pertaining to the number of the long-term co-integration.

**Pattern 1:** $H_0$ is the maximal number of vectors indicating the long-term co-integration equivalent to $r$. $H_1$ is the number of vectors indicating the long-term co-integration greater than $r$. In the above,

$r = 0,1,2,\ldots,n-1$; the statistical value to testify the above assumption is trace statistic $(\lambda_{trace})$, which can be computed using the equation below:

$$\lambda_{trace}(r) = -T \sum_{i=r+1}^{n} (1 - \hat{\lambda}_i) \tag{14}$$

**Pattern 2:** $H_0$ is the maximal number of vectors indicating the long-term co-integration equivalent to $r$. $H_1$ is the number of vectors indicating the long-term co-integration equivalent to $r + 1$. In the above, $r = 0, 1, 2, \ldots, n-1$, and the statistical value to testify the above assumption is maximum eigenvalue $\lambda_{\max}$, which can be computed using the equation below:

$$\lambda_{\max}(r, r+1) = -T(1 - \hat{\lambda}_{r+1}) \tag{15}$$

$$\hat{A}_i = \begin{cases} I + \widehat{\Pi} + \widehat{\Gamma} & ,i = 1 \\ \widehat{\Gamma}_i - \widehat{\Gamma}_{i-1} & ,2 \leq i \leq -1 \\ -\widehat{\Gamma}_{p-1} & i = p \end{cases} \tag{16}$$

After that, we used the SEM-VECM forecasting model of the time series in vector $X_t$ by using the same concept, which is the forecasting of the minimum mean square error. Hence, the forecast of $1, 2, \ldots, h$ pre-timing of the time series in the vector $X_t$ can be illustrated as:

$$\hat{X}_{T+1} = \hat{A}_1 X_T + \hat{A}_2 X_{T-1} + \hat{A}_p X_{T-p+1} \tag{17}$$

$$\hat{X}_{T+2} = \hat{A}_1 X_{T+1} + \hat{A}_2 X_{T-1} + \hat{A}_p X_{T-p+2} \tag{18}$$

$$\hat{X}_{T+h} = \hat{A}_1 X_{T+h-1} + \hat{A}_2 X_{T+h-2} + \ldots + \hat{A}_p X_{T-p+h} \tag{19}$$

where $\hat{X}_{T+j} = \hat{A}_1 X_{T+j}$ if $j < 0$.

*2.2. Measurement of the Forecasting Performance*

For this research, we decided to use the mean absolute percentage error (MAPE) and root mean square error (RMSE) to compare the forecasting accuracy of each model [39,40]. The calculation equations are shown as follows:

$$MAPE = \frac{1}{n} \sum_{i=1}^{n} \left| \frac{\hat{y}_i - y_i}{y_i} \right| \tag{20}$$

$$RMSE = \sqrt{\frac{1}{n} \sum_{i=1}^{n} (\hat{y}_i - y_i)^2} \tag{21}$$

## 3. Empirical Analysis

*3.1. Screening of Influencing Factors for Model Input*

This paper identified a structure equation modeling containing both exogenous variables and endogenous variables, in which the variables are both latent variables and observed variables. The latent variables comprise economic growth, social growth, and environmental growth, whereas the observed variables include nine factors: carbon dioxide emissions ($CO_2$), per capita GDP (GDP), labor growth(L), urbanization rate (UR), industrial structure (IS), energy consumption (EC), foreign direct investment (FDI), oil price (OP), and net exports $(X - E)$. However, all causal factors we used in the structure equation modeling were confirmed to be stationary at the same level according to the Augmented Dickey-Fuller theory. Therefore, this study found that all nine causal factors are stationary at the First Difference I(1), as illustrated in Table 1.

**Table 1.** Augment Dickey Fuller (ADF) test at First Difference I(1).

| ADF Test at First Difference I(1) | | MacKinnon Critical Value | | |
|---|---|---|---|---|
| **Variables** | **Value** | **1%** | **5%** | **10%** |
| $\Delta\ln(CO_2)$ | −6.79 *** | −4.75 | −3.41 | −2.77 |
| $\Delta\ln(GDP)$ | −5.92 *** | −4.75 | −3.41 | −2.77 |
| $\Delta\ln(L)$ | −4.77 *** | −4.75 | −3.41 | −2.77 |
| $\Delta\ln(UR)$ | −6.51 *** | −4.75 | −3.41 | −2.77 |
| $\Delta\ln(IS)$ | −5.99 *** | −4.75 | −3.41 | −2.77 |
| $\Delta\ln(EC)$ | −6.47 *** | −4.75 | −3.41 | −2.77 |
| $\Delta\ln(FDI)$ | −4.99 *** | −4.75 | −3.41 | −2.77 |
| $\Delta\ln(OP)$ | −6.54 *** | −4.75 | −3.41 | −2.77 |
| $\Delta\ln(X-E)$ | −6.13 *** | −4.75 | −3.41 | −2.77 |

**Note.** $CO_2$ is the carbon dioxide emissions; GDP is the per capita GDP; $L$ is the labor growth; $UR$ is the urbanization rate, $IS$ is the industrial structure, $EC$ is the energy consumption, $FDI$ is the foreign direct investment, $OP$ is the oil price, and $X-E$ is net exports, *** denotes a significance, $\alpha = 0.01$, compared to the Tau test with the MacKinnon Critical Value, $\Delta$ is the first difference, and ln is the natural logarithm.

In this paper, we have defined the structure equation modeling with exogenous variables and endogenous variables for latent variables, such as economic growth, social growth, and environmental growth, and the observed variables are nine factors: carbon dioxide emissions, per capita GDP, labor growth, urbanization rate, industrial structure, energy consumption, foreign direct investment, oil price, and net exports. However, we have determined that all causal factors analyzed in the structural equation modeling must be stationary at the same level according to the Augmented Dickey-Fuller theory. The nine causes are stationary at the First Difference I(1), We deployed a time-series data from 1977 to 1990 in the analysis. In addition, we tested the estimated value as to whether it is not spurious or otherwise by assessing the problem of heteroskedasticity, perfect multicollinearity, and autocorrelation, as illustrated in Table 1.

Table 1 demonstrates that all factors are non-stationary at Level I(0). The factors were carried out for a first difference: it was found that those factors became stationary at Level I(1). Here, the value of the ADF test was greater than the MacKinnon Critical Value. This shows that the factors were at a statistical significance level of 1%, 5%, and 10%. Once the factors were identified as stationary, they were taken for a co-integration test, as suggested by Johansen Juselius as shown in Table 2.

### 3.2. Analysis of Co-Integration

Table 2 presents the results of co-integration test. The results verified that each causal factor was co-integrated at a confidence interval of 99%. This is because the trace test results were 241.65 and 89.15, whereas the maximum eigenvalue test results were 135.09 and 91.50, which are higher than MacKinnon Critical Values at a significance level of 1% and 5%. In addition, the factors were used to construct the SEM-VECM Model.

**Table 2.** Co-integration test by Johansen Juselius.

| Variables | Hypothesized No of CE(S) | Trace Statistic Test | Max-Eigen Statistic Test | MacKinnon Critical Value | |
|---|---|---|---|---|---|
| | | | | **1%** | **5%** |
| $\Delta\ln(CO_2)$, $\Delta\ln(GDP)$, $\Delta\ln(L)$, $\Delta\ln(UR)$, $\Delta\ln(IS)$, | None *** | 241.65 | 135.09 | 25.25 | 12.50 |
| $\Delta\ln(EC)$, $\Delta\ln(FDI)$, $\Delta\ln(OP)$, $\Delta\ln(X-E)$ | At Most 1 *** | 89.15 | 91.50 | 5.60 | 3.50 |

*** denotes significance $\alpha = 0.01$.

### 3.3. Formation of Analysis Modeling with the SEM-VECM Model

The SEM-VECM Model is a model optimized to illustrate the relationship of causal factors in both the short and long term. The direct and indirect effect relationships of factors are shown in Figure 2. In addition, the paper examines the model with regards to heteroscedasticity, multicollinearity and

autocorrelation. The examination results show that the SEM-VECM Model is free from those three issues. The analysis results of the relationships demonstrated by the SEM-VECM Model are presented in Table 3.

**Table 3.** Results of relationship size analysis of the SEM-VECM Model.

| Dependent Variables | Type of Effect | Independent Variables | | | | | | | | |
|---|---|---|---|---|---|---|---|---|---|---|
| | | $\Delta\ln(GDP)_{t-1}$ | $\Delta\ln(L)_{t-1}$ | $\Delta\ln(UR)_{t-1}$ | $\Delta\ln(IS)_{t-1}$ | $\Delta\ln(EC)_{t-1}$ | $\Delta\ln(FDI)_{t-1}$ | $\Delta\ln(OP)_{t-1}$ | $\Delta\ln(X-E)_{t-1}$ | $ECM_{t-1}$ |
| $\Delta\ln(CO_2)_{t-1}$ | DE | 0.82 *** | 0.31 ** | 0.69 *** | 0.74 *** | 0.80 *** | 0.66 ** | 0.47 ** | 0.35 *** | 0.39 *** |
| | IE | 0.11 *** | 0.15 ** | 0.04 *** | 0.09 *** | 0.02 *** | 0.01 ** | 0.05 ** | 0.11 *** | - |
| $\Delta\ln(GDP)_{t-1}$ | DE | - | 0.09 *** | 0.64 *** | 0.36 *** | 0.55 *** | 0.32 *** | - | 0.73 *** | - |
| | IE | - | - | - | - | - | - | - | - | - |
| $\Delta\ln(L)_{t-1}$ | DE | - | - | 0.62*** | - | - | - | - | - | - |
| | IE | - | - | - | - | - | - | - | - | - |
| $\Delta\ln(UR)_{t-1}$ | DE | - | - | - | - | - | - | - | - | - |
| | IE | - | - | - | - | - | - | - | - | - |
| $\Delta\ln(IS)_{t-1}$ | DE | - | - | - | - | - | - | 0.22 *** | - | - |
| | IE | - | - | - | - | - | - | - | - | - |
| $\Delta\ln(EC)_{t-1}$ | DE | - | 0.45 *** | - | - | - | - | 0.43 *** | - | - |
| | IE | - | - | - | - | - | - | - | - | - |
| $\Delta\ln(FDI)_{t-1}$ | DE | - | - | - | - | - | - | - | - | - |
| | IE | - | - | - | - | - | - | - | - | - |
| $\Delta\ln(OP)_{t-1}$ | DE | - | - | - | - | - | - | - | - | - |
| | IE | - | - | - | - | - | - | - | - | - |
| $\Delta\ln(X-E)_{t-1}$ | DE | - | - | 0.49 *** | - | - | - | 0.41 ** | - | - |
| | IE | - | - | - | - | - | - | - | - | - |

Note: In the above, $ECM_{t-1}$ is Error Correction Mechanism test, *** denotes significance $\alpha = 0.01$, ** denotes significance $\alpha = 0.05$, $\chi^2/df$ is 1.19, root mean square of error approximation ($RMSEA$) is 0.02, root mean squared residual ($RMR$) is 0.008, goodness of fit index ($GFI$) is 0.97, adjusted goodness of fit index ($AGFI$) is 0.99, R-squared is 0.95, adjusted R-squared is 0.94, the Durbin-Watson statistic is 1.89, the F-statistic is 275.00 (probability is 0.00), the ARCH test is 35.01 (probability is 0.1), the LM test is 1.36 (probability is 0.10), DE is direct effect and IE is indirect effect.

Figure 3 demonstrates the analysis results of relationship of the causal factors in the SEM-VECM Model. The factors of the model consist of $\Delta\ln(CO_2)_{t-1}$, $\Delta\ln(GDP)_{t-1}$, $\Delta\ln(L)_{t-1}$, $\Delta\ln(UR)_{t-1}$, $\Delta\ln(IS)_{t-1}$, $\Delta\ln(EC)_{t-1}$, $\Delta\ln(FDI)_{t-1}$, $\Delta\ln(OP)_{t-1}$, $\Delta\ln(X-E)_{t-1}$ and $ECM_{t-1}$. The study indicates that each causal factor has both direct and indirect effect relationships, while the relationship size can be found in Table 3.

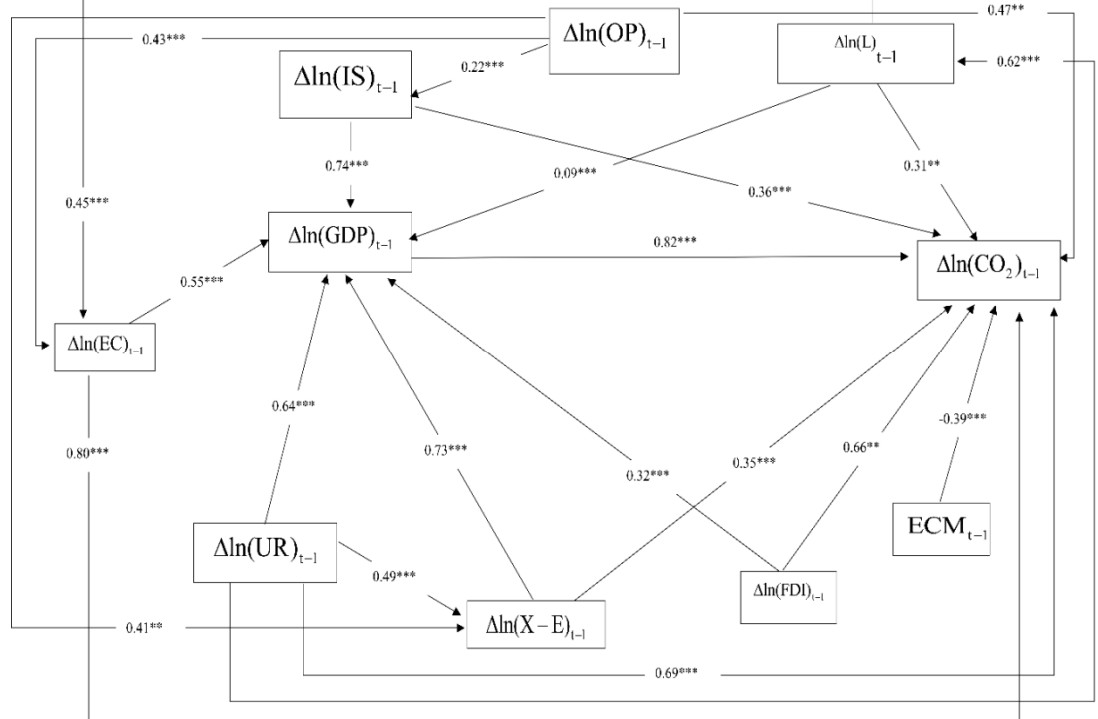

**Figure 3.** Relationships of causal factors in the SEM-VECM Model.

Table 3 illustrates the parameters of the SEM-VECM Model with a statistical significance level of 1% and 5%. The study analysis confirms that each causal factor has an influence on changes in $CO_2$ emissions in both direct effect and indirect effect, as can be observed in the following. The factor of per capita GDP ($\Delta\ln(\text{GDP})_{t-1}$) had a direct effect on $CO_2$ emissions ($\Delta\ln(CO_2)_{t-1}$), equivalent to 82.00%, and its indirect effect was equal to 11.00% with a significance level of 1%. Labor growth ($\Delta\ln(L)_{t-1}$) had a direct effect on $CO_2$ emissions ($\Delta\ln(CO_2)_{t-1}$) at 31.00%, while its indirect effect was equal to 15.00% with a significance level of 5%. In addition, labor growth ($\Delta\ln(L)_{t-1}$) had a direct effect on per capita GDP ($\Delta\ln(\text{GDP})_{t-1}$) equivalent to 9.00% with a significance level of 1%, and it also ($\Delta\ln(L)_{t-1}$) had a direct effect on energy consumption ($\Delta\ln(EC)_{t-1}$) equivalent to 45.00% with a significance level of 1%. The urbanization rate factor ($\Delta\ln(URT)_{t-1}$) had a direct effect on $CO_2$ emissions ($\Delta\ln(CO_2)_{t-1}$) equivalent to 69.00%, and its indirect effect was equal to 4.00% with a significance level of 1%. Furthermore, urbanization rate ($\Delta\ln(URT)_{t-1}$) had a direct effect on per capita GDP ($\Delta\ln(\text{GDP})_{t-1}$) equivalent to 64.00% with a significance level of 1%. Urbanization rate ($\Delta\ln(URT)_{t-1}$) had a direct effect on labor growth ($\Delta\ln(L)_{t-1}$) equivalent to 62.00% with a significance level of 1%, and the urbanization rate ($\Delta\ln(URT)_{t-1}$) also had a direct effect on net exports ($\Delta\ln(X-E)_{t-1}$) equivalent to 49.00% with a significance level of 1%. As for industrial structure ($\Delta\ln(IS)_{t-1}$), it had a direct effect on $CO_2$ emissions ($\Delta\ln(CO_2)_{t-1}$) equivalent to 74.00% and its indirect effect was equal to 9.00% with a significance level of 1%. In addition, industrial structure ($\Delta\ln(IS)_{t-1}$) had a direct effect on per capita GDP ($\Delta\ln(\text{GDP})_{t-1}$) equivalent to 36.00% with a significance level of 1%. Energy consumption ($\Delta\ln(EC)_{t-1}$) had a direct effect on $CO_2$ emissions ($\Delta\ln(CO_2)_{t-1}$) at 80%, and its indirect effect was equal to 2% with a significance level of 1%. With regards to energy consumption ($\Delta\ln(EC)_{t-1}$), it had a direct effect on per capita GDP ($\Delta\ln(\text{GDP})_{t-1}$) at 55% with a significance level of 1%. Foreign direct investment ($\Delta\ln(FDI)_{t-1}$) had a direct effect on $CO_2$ emissions ($\Delta\ln(CO_2)_{t-1}$) at 66%, and its indirect effect was equal to 1% with a significance level of 1%. Furthermore, foreign direct investment ($\Delta\ln(FDI)_{t-1}$) had a direct effect on per capita GDP ($\Delta\ln(\text{GDP})_{t-1}$) at 32% with a significance level of 1%. Oil price ($\Delta\ln(OP)_{t-1}$) had a direct effect on $CO_2$ emissions ($\Delta\ln(CO_2)_{t-1}$) equivalent to 47%, and its indirect effect is equal to 5% with a significance level of 5%. Oil price ($\Delta\ln(OP)_{t-1}$) had a direct effect on industrial structure ($\Delta\ln(IS)_{t-1}$) at 22% with a significance level of 1%, while oil price ($\Delta\ln(OP)_{t-1}$) had a direct effect on energy consumption ($\Delta\ln(EC)_{t-1}$) at about 43% with a significance level of 1%. Moreover, oil price ($\Delta\ln(OP)_{t-1}$) had a direct effect on net exports ($\Delta\ln(X-E)_{t-1}$) at about 41% with a significance level of 5%. Net exports ($\Delta\ln(X-E)_{t-1}$) had a direct effect on $CO_2$ emissions ($\Delta\ln(CO_2)_{t-1}$) at about 35%, and its indirect effect was about 11% with a significance level of 1%. In addition, net exports ($\Delta\ln(X-E)_{t-1}$) had a direct effect on per capita GDP ($\Delta\ln(\text{GDP})_{t-1}$) at 73% with a significance level of 1%.

$ECM_{t-1}$ had a parameter of $-0.39$, which indicates that the adjustment ability of the SEM-VECM Model to the equilibrium is at the rate of 39%.

Based on the analysis shown in Table 3, all variables were included in the forecast. In order to see the effectiveness of the SEM-VECM model, we conducted a comparison of forecasting performance by using the MAPE and RMSE. This comparison was paired against other models, namely, the multiple linear regression, artificial neural natural (ANN), back propagation neural network (BP Model), gray (GM (1,1)), ARIMA and GM-ARIMA models, as demonstrated below.

Table 4 shows that the SEM-VECM Model had the lowest value of MAPE and RMSE equivalent to 1.21% and 1.02%, respectively. In comparison with other models, this produces the following results. The GM-ARIMA Model had a MAPE and RMSE of 4.63% and 4.09%, respectively. The ARIMA Model presented a MAPE and RMSE with a percentage of 4.97 and 6.07, respectively. The Gray Model (GM (1,1) had a MAPE and RMSE of 8.61% and 7.98%, respectively. The Back propagation neural network (BP Model) had a MAPE and RMSE of 10.15% and 10.03%, respectively. The Artificial Neural Natural Model (ANN) generated a MAPE and RMSE of 15.54% and 14.22%, respectively, while the Multiple Linear Regression model had a MAPE and RMSE equivalent to 23.09% and 21.39%,

respectively. Thus, the SEM-VECM Model is seen to be the most appropriate model for the future long-term forecasting.

**Table 4.** The performance monitoring of the forecasting model.

| Forecasting Model | Mean Absolute Percentage Error (MAPE) (%) | Root Mean Square Error (RMSE) (%) |
|---|---|---|
| Multiple Linear Regression model | 23.09 | 21.39 |
| Artificial Neural Natural Model (ANN) | 15.54 | 14.22 |
| Back propagation neural network (BP model) | 10.15 | 10.03 |
| Gray model (GM (1,1)) | 8.61 | 7.98 |
| ARIMA model | 4.97 | 6.07 |
| GM-ARIMA Model | 4.63 | 4.09 |
| SEM-VECM Model | 1.21 | 1.02 |

*3.4. $CO_2$ Emission Forecasting Based on the SEM-VECM Model*

We later employed the SEM-VECM Model to predict changes in $CO_2$ emissions in Thailand's transportation sector for the next 30 years (2018–2047), as shown in Figure 4.

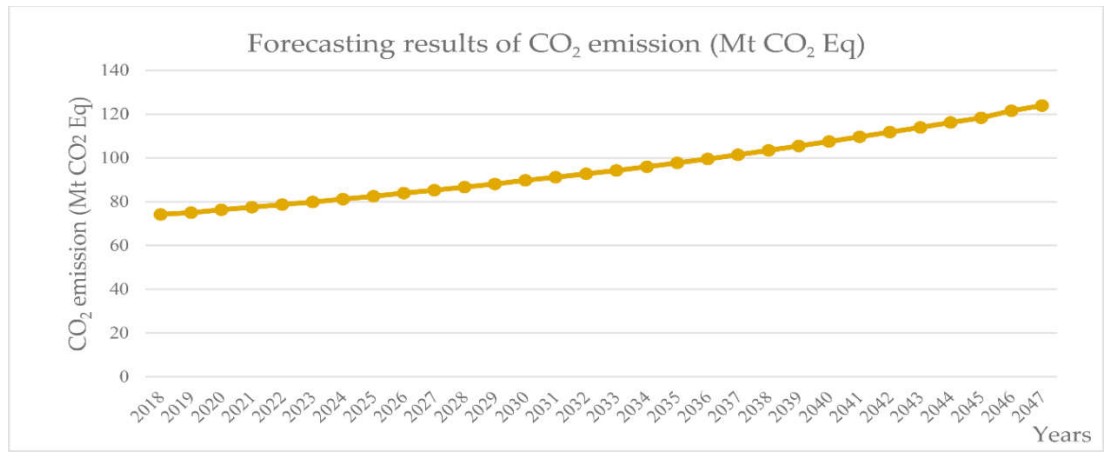

**Figure 4.** Forecasting results of $CO_2$ emissions from 2018 to 2047 in Thailand's transportation sector.

Figure 4 shows that $CO_2$ emissions in Thailand's transportation sector over the next 30 years (2018 to 2047) are projected to increase with a growth rate of 67.04%, producing a significant amount of $CO_2$ emissions, continuously adding to the greenhouse gas problem.

## 4. Discussion

This research was conducted to develop a new long-term forecasting model adapted from existing, older models. The newly-developed SEM-VECM Model was designed for use in long-term forecasting of $CO_2$ emissions in Thailand's transportation sector over the next 30 years (2018–2047). The SEM-VECM Model was constructed utilizing various relevant forecasting theories. The study found that the SEM-VECM Model is the most efficient model with the lowest MAPE and RMSE as compared to the other models, namely GM-ARIMA Model, ARIMA Model, Gray Model (GM (1,1)), Back propagation neural network (BP Model), Artificial Neural Natural Model (ANN), and Multiple Linear Regression Model.

The SEM-VECM Model projected $CO_2$ emissions from the Thai transportation sector to increase steadily over the projection period. This indicates that $CO_2$ levels will likely increase beyond projections used for the government's current management strategy. At the same time, the results reflect that the current management plans and approaches will likely not be effective enough to achieve its sustainable development goals. Therefore, the responsible parties must focus on the formulation of new policies

and management strategies taking into account each causal factor affecting changes in $CO_2$ emissions in the transportation sector, be it a direct and indirect effect. Otherwise, poor planning could result in serious negative consequences for both the economy and environment.

This research took a unique approach to forecasting $CO_2$ emissions from the transportation sector. It was structured based on an analysis of the causal factors, optimizing the Structural Equation Model, and utilizing the VECM Model, which had not been done in previous studies. The researchers chose to use LISREL software integrating Microsoft office to produce the most efficient and effective tool, which is also suitable for application in other sectors. The SEM-VECM Model passed an analysis of co-integration test and error correction mechanism test, which testifies to its ability as an ideal model for checking heteroskedasticity, multicollinearity, and autocorrelation. Throughout the study, it can be observed that the SEM-VECM Model is capable of predicting long-term changes more effectively compared to older models.

This paper shares similarities with past studies with regards to predicting $CO_2$ emissions in both the short and long term by sector. However, few studies forecast more than 20 years into the future. The current study integrated advanced statistics in its modeling and made special effort to minimize errors. However, since the model deals with long-term prediction, there are possible factors that could affect its accuracy. Therefore, the paper emphasized the research processes, and used only the specified causal factors with strict criteria in the selection process. If certain variables did not meet the criteria, they were immediately removed from the model. This is another factor which distinguishes this model from other existing forecasting models.

With regards to further study, any interested individuals should pay careful attention to choosing causal factors and the research process in reference to this research. Additionally, the estimation of parameters must use advanced statistics to close research gaps of past models in developing new models. Causal factors must be stationary at the same level, and adjustment to the equilibrium must be taken into a consideration. Furthermore, the Error Correction Mechanism should always be included in the model.

A limitation of this research is that Thailand's current sustainable development policy does not take into account new causal factors. This will no doubt change in the future. One new factor that should be considered is how the continued promotion of national tourism will contribute to future changes in $CO_2$ emissions. It is recommended that the government reviews and revises the relevant current policies based effective implementation of this forecasting model so as to increase managerial efficiency and effectiveness.

## 5. Conclusions

As part of the contributions of this paper, we have established a forecasting model called the "SEM-VECM Model," which is an application of the SEM model that incorporates the VECM Model. This application was initiated in order to increase the efficiency and effectiveness of the established model. Additionally, it attempts to pave a guideline for future research. In the modeling of the SEM-VECM Model, only stationary variables at the first difference were selected. They included per capita GDP ($\Delta\ln(GDP)_{t-1}$), labor growth ($\Delta\ln(L)_{t-1}$), urbanization rate factor ($\Delta\ln(URT)_{t-1}$), industrial structure ($\Delta\ln(IS)_{t-1}$), energy consumption ($\Delta\ln(EC)_{t-1}$), foreign direct investment ($\Delta\ln(FDI)_{t-1}$), oil price ($\Delta\ln(OP)_{t-1}$), and net exports ($\Delta\ln(X-E)_{t-1}$). Those variables then underwent a co-integration test. The study showed that all variables are co-integrated, and this enabled us to include ECM into the analysis of the SEM-VECM model.

As for the study's findings, it is evident that the SEM-VECM model can adjust itself toward the equilibrium with a ratio of 39%, and this implies that all variables have a direct effect on the changes of $CO_2$ emissions during the time period ($t-1$) at confidence intervals of 99% and 95%. In addition, all factors were found to have an indirect effect on the change of $CO_2$ emissions during the time period ($t-1$) at confidence intervals of 99% and 95%, except for the error correction mechanism (ECM) variable. Additionally, the study reveals that all variables are influential over one variable to another,

with both direct and indirect effects at confidence intervals of 99% and 95%. Therefore, the use of the SEM-VECM Model for a $CO_2$ emissions prediction is more effective when compared to previously studied models.

As for the study's results, it will have effective applications for future research and in different research contexts or sectors. Moreover, it will help to facilitate national policy planning in the future. Nonetheless, for maximal application of this research in the future, any interested individual will have to develop a new model by adapting the established model, while other concepts need to be taken into account. For instance, the concept of path analysis along with the Vector Autocorrective (VAR) model can be used for modeling, or the factor analysis with the Autoregressive Integrated Moving Average with Exogenous Variables (ARIMAX) model should be adapted to construct a model. All of these different applications of various concepts are to produce a useful output to achieve sustainable development goals.

**Author Contributions:** P.S. and D.A. were involved in the data collection and preprocessing phase, model constructing, empirical research, results analysis and discussion, and manuscript preparation. All authors have approved the submitted manuscript.

**Funding:** This research received no external funding.

**Acknowledgments:** This research has been supported by the Rachadapisek Sompote Fund for Postdoctoral Fellowship, Chulalongkorn University.

**Conflicts of Interest:** The authors declare no conflict of interest.

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
