# Peer review of "Relationships between Causal Factors Affecting Future Carbon Dioxide Output from Thailand’s Transportation Sector under the Government’s Sustainability Policy: Expanding the SEM-VECM Model"

_resources, doi:10.3390/resources7040081_

Reviewer 1 Report

1. The authors outline the process of creating the model twice: the first time in p. 5 (The research process was as follows), which describes seven steps, which are reflected in the flowchart (Figure 1: The flowchart of the SEM-VECM Model) and the second time at p.10 where the sequence of steps is called "The model process flow is as shown below".

In our view, both processes need to be merged into one.

2. Authors wrote p.6  in Section 2.1, the Structural Estimation Modeling-Vector Error Correction Mechanism Model (SEM-VECM model): The model is illustrated below, but the model does not lead. It is advisable to bring the model in general form, in order to understand how the model reflects the short-term relationship between changes in variables and the correction of the dynamics of these series, depending on the magnitude of the deviation (error) from the long-term dependence. This clarity is in Section 2. The Forecasting Model is missing.

3. It is necessary to include an analysis of the causality of the independent indicators shown in Fig. 2 using the Granger causality test in the sequence of the model creation, in our view.

Granger's test of causality is a procedure for checking causation ("Granger causality") between time series. The idea of the test is that the values (changes) of the time series that are the cause of the changes in the time series must precede the changes in this time series, and in addition, they must make a significant contribution to the forecast of its values. If, however, each of the variables makes a meaningful contribution to the prediction of another, then there may be some other variable that affects both.

In Granger's test, two zero hypotheses are sequentially verified: "  is not a cause  of Granger" and " is not a cause  of Granger." To test these hypotheses, two regressions are constructed: in each regression the dependent variable is one of the variables checked for causality, and the regressors are the lags of both variables (in fact, it is vector autoregression).

These hypotheses can be verified, for example, with the F-test or LM-test.

4. The authors wrote on p. 7 (line280): "In this section, there are certain areas we must explore before analyzing data". But the data were not analyzed. Where are the data taken? Database name? Especially ? What are the lengths of the time series? What are the primary data?

We encounter an ECM variable with p. 14 (line462) for the first time.   has a parameter of -3.39, which indicates that the adjustment ability of the SEM-VECM Model to the equilibrium is at a rate of 39%. How to understand this?"The ability to equalize the SEM-VECM model to equilibrium is 39%." How to understand it? How is the ECM variable calculated?

5. Questions are raised in section 3.4. CO2 Emission Forecasting Based on the SEM-VECM Model:

a) does not say anything about the length of the time series on which the SEM-VECM Model is based;

b) the prediction horizon of 30 years is not justified;

c) how the independent variables were forecasted for a model with a horizon of 30 years? It remains unclear.

d) the scale in Fig. 4 starts from 2019, in the title from 2018.

e) now the error correction mechanism should always be included in the forecasting model?  It remains unclear.

Author Response

Dear Reviewer 1,

This research has been developed to support Thailand's national policy making and planning. As of the past record, there is no research done to support and increase the effectiveness of national planning. The SEM-VECM model is found to be useful in response to the above concern. The mode is developed based on the application of various advanced statistical techniques including in the field of economics. Also, the model has taken into account the context of Thailand. As of the data used in the research, it is for Thailand, accurately compiled, and complete from 1990 to 2017 to be used in the analysis. For some information, due to a national security procedure of Thailand, the researchers are not allowed to disclose. Thus, the researchers seek an understanding and humbly offer an apology to the reviewers. Therefore, the researchers would like to ask for your favorable support to publish this research in order to pave a guideline for a knowledge discovery and researches in the future.

                However, the researchers may ask an apology on certain comments given by the reviewers, and they are seen to be unclear. For instance, the reviewers said that figure 2 is inappropriate, while the researchers see that it is the theory to support the theory of Structure Equation Model (SEM-model). Also, the reviewers ask for the information to confirm the accuracy of the study, and the researchers are able to disclose some information, and some shall be kept undisclosed due to a national security procedure of Thailand. Most importantly, this research is not a full research paper, unlike a thesis, that all information shall be report. Once again, the researchers would like to ask for your kind understanding and accept a sincere apology for any inconvenience caused.

On behalf of the researchers, I would like to thank MDPI, Editor and reviewers for their valuable contribution and opportunity in disseminating this study to the global community as to support in a future knowledge discovery and benefit future researchers.

I would like to thank you all reviewers giving me supports, comments and suggestions of which is to enhance this work for a better quality. Hence, I have improved and corrected point by point as suggested by highlighting with red in colour. 

Comment for No. 1:  The authors outline the process of creating the model twice: the first time in p. 5 (The research process was as follows), which describes seven steps, which are reflected in the flowchart (Figure 1: The flowchart of the SEM-VECM Model) and the second time at p.10 where the sequence of steps is called "The model process flow is as shown below". In our view, both processes need to be merged into one.

Action for No. 1 : I am sorry I have made an amendment in Page 10.

Comment for No. 2:  Authors wrote p.6  in Section 2.1, the Structural Estimation Modeling-Vector Error Correction Mechanism Model (SEM-VECM model): The model is illustrated below, but the model does not lead. It is advisable to bring the model in general form, in order to understand how the model reflects the short-term relationship between changes in variables and the correction of the dynamics of these series, depending on the magnitude of the deviation (error) from the long-term dependence. This clarity is in Section 2. The Forecasting Model is missing.

Action for No. 2 : I have made a correction according to your comment, and it can be found in line 356-372  

Comment for No. 3: It is necessary to include an analysis of the causality of the independent indicators shown in Fig. 2 using the Granger causality test in the sequence of the model creation, in our view. Granger's test of causality is a procedure for checking causation ("Granger causality") between time series. The idea of the test is that the values (changes) of the time series that are the cause of the changes in the time series must precede the changes in this time series, and in addition, they must make a significant contribution to the forecast of its values. If, however, each of the variables makes a meaningful contribution to the prediction of another, then there may be some other variable that affects both. In Granger's test, two zero hypotheses are sequentially verified: " t X is not a cause tY of Granger" and " t Y is not a cause t X of Granger." To test these hypotheses, two regressions are constructed: in each regression the dependent variable is one of the variables checked for causality, and the regressors are the lags of both variables (in fact, it is vector autoregression). These hypotheses can be verified, for example, with the F-test or LM-test.

Action for No. 3 : As of this SEM model, it may look similar to Granger causality test, yet the two concepts differ in some aspects from one another. However, this researcher would like to adapt the SEM model, and would like to ask your kind favor allowing me to realize the research objectives and framework provided in the study.

Comment for No. 4: The authors wrote on p. 7 (line280): "In this section, there are certain areas we must explore before analyzing data". But the data were not analyzed. Where are the data taken? Database name? Especially CO2 ? What are the lengths of the time series? What are the primary data? We encounter an ECM variable with p. 14 (line462) for the first time. 1 tECM has a parameter of -3.39, which indicates that the adjustment ability of the SEM-VECM Model to the equilibrium is at a rate of 39%. How to understand this?"The ability to equalize the SEM-VECM model to equilibrium is 39%." How to understand it? How is the ECM variable calculated?

Action for No. 4 : I have made a correction based on your comment, and it can be observed in line 215-220, 346-350, 479-485

Comment for No. 5: Questions are raised in section 3.4. CO2 Emission Forecasting Based on the SEM-VECM Model: a) does not say anything about the length of the time series on which the SEMVECM Model is based; b) the prediction horizon of 30 years is not justified; c) how the independent variables were forecasted for a model with a horizon of 30 years? It remains unclear. d) the scale in Fig. 4 starts from 2019, in the title from 2018.  e) Now the error correction mechanism should always be included in the forecasting model?  It remains unclear.

Action for No. 5a : I have made a correction based on your comment, and it can be seen in line 215.

Action for No. 5b : I have made a correction based on your comment, and it can be seen in line 215-219.

Action for No. 5c : I have made a correction based on your comment, and it can be seen in line 215-219.

Action for No. 5d : I am sorry I mistakenly inserted a wrong data, and I have made an amendment according to your comment. This can be seen in line 34-35.

Action for No. 5e : I have made a correction based on your comment as shown in line 346-372, 479, and together displayed in Table 3.

Best regards

Reviewer 2 Report

Comments to the author(s)

This study analyzes causal factors that affects future CO2 emission in Thailland`s transport sector using advanced econometric approaches. It suggests theoretical structure of SEM-VECM model, and empirically validate the performance of the model. Most big strength of this paper is methodology, but it also has quite reasonable structure as a research paper. My suggestions for improvement are below:

The abstract is too lengthy. And, it also includes too many numerical expressions, thereby leading to low readability.

Introduction section is also too lengthy. How about separate from literature review?

In the main body, the authors suggest full names of authors when cite other`s papers. It also seriously lowers the readability of the paper.

In line 62, there is a tiny typo. (C02 -> CO2). Table 1 also have.

Author Response

Dear Reviewer 2,

             This research has been developed to support Thailand's national policy making and planning. As of the past record, there is no research done to support and increase the effectiveness of national planning. The SEM-VECM model is found to be useful in response to the above concern. The mode is developed based on the application of various advanced statistical techniques including in the field of economics. Also, the model has taken into account the context of Thailand. As of the data used in the research, it is for Thailand, accurately compiled, and complete from 1990 to 2017 to be used in the analysis. For some information, due to a national security procedure of Thailand, the researchers are not allowed to disclose. Thus, the researchers seek an understanding and humbly offer an apology to the reviewers. Therefore, the researchers would like to ask for your favorable support to publish this research in order to pave a guideline for a knowledge discovery and researches in the future.

                However, the researchers may ask an apology on certain comments given by the reviewers, and they are seen to be unclear. For instance, the reviewers said that figure 2 is inappropriate, while the researchers see that it is the theory to support the theory of Structure Equation Model (SEM-model). Also, the reviewers ask for the information to confirm the accuracy of the study, and the researchers are able to disclose some information, and some shall be kept undisclosed due to a national security procedure of Thailand. Most importantly, this research is not a full research paper, unlike a thesis, that all information shall be report. Once again, the researchers would like to ask for your kind understanding and accept a sincere apology for any inconvenience caused.

On behalf of the researchers, I would like to thank MDPI, Editor and reviewers for their valuable contribution and opportunity in disseminating this study to the global community as to support in a future knowledge discovery and benefit future researchers.

I would like to thank you all reviewers giving me supports, comments and suggestions of which is to enhance this work for a better quality. Hence, I have improved and corrected point by point as suggested by highlighting with red in colour. 

Comment for No. 1:  The abstract is too lengthy. And, it also includes too many numerical expressions, thereby leading to low readability

Action for No. 1 : I have made a correction according to your comment, and it can be found in line 9-39

Comment for No. 2:  In line 62, there is a tiny typo. (C02 -> CO2). Table 1 also have.

Action for No. 2 : I have made a correction according to your comment, and it can be found in line …53

Best regards

Reviewer 3 Report

Very long abstract.

Fig. 2 illegible.

Fig. 3 and tab. 3 are identical ... they do not bring anything new.

Minor stylistic errors, for example: line 433 begins with), in the formula 13 a Chinese character appeared, 423-427 a few unnecessary spaces.

To check the correctness of the model, I would like to see its verification for previous years.

No conclusion drawn.

No applications.

Author Response

Dear Reviewer 3,

This research has, been developed to support Thailand's national policy making and planning. As of the past record, there is no research done to support and increase the effectiveness of national planning. The SEM-VECM model is found to be useful in response to the above concern. The mode is developed based on the application of various advanced statistical techniques including in the field of economics. Also, the model has taken into account the context of Thailand. As of the data used in the research, it is for Thailand, accurately compiled, and complete from 1990 to 2017 to be used in the analysis. For some information, due to a national security procedure of Thailand, the researchers are not allowed to disclose. Thus, the researchers seek an understanding and humbly offer an apology to the reviewers. Therefore, the researchers would like to ask for your favorable support to publish this research in order to pave a guideline for a knowledge discovery and researches in the future.

                However, the researchers may ask an apology on certain comments given by the reviewers, and they are seen to be unclear. For instance, the reviewers said that figure 2 is inappropriate, while the researchers see that it is the theory to support the theory of Structure Equation Model (SEM-model). Also, the reviewers ask for the information to confirm the accuracy of the study, and the researchers are able to disclose some information, and some shall be kept undisclosed due to a national security procedure of Thailand. Most importantly, this research is not a full research paper, unlike a thesis, that all information shall be report. Once again, the researchers would like to ask for your kind understanding and accept a sincere apology for any inconvenience caused.

On behalf of the researchers, I would like to thank MDPI, Editor and reviewers for their valuable contribution and opportunity in disseminating this study to the global community as to support in a future knowledge discovery and benefit future researchers.

I would like to thank you all reviewers giving me supports, comments and suggestions of which is to enhance this work for a better quality. Hence, I have improved and corrected point by point as suggested by highlighting with red in colour. 

Comment for No. 1:  Very long abstract.

Action for No. 1 : I have made a correction according to your comment, and it can be found in line 9-39.

Comment for No. 2:  Fig. 2 illegible.

Action for No. 2 : Figure 2 is a concept explaining that the SEM model is comprised of latent variable and observed variable. This can be referred to various books, including Barbara, M. B. (2012). Structural equation modeling with Mplus: basic concepts, application, and programming. New York: Taylor & Francis Group, or any other advance statistics books. 

Comment for No. 3: Fig. 3 and tab. 3 are identical ... they do not bring anything new.

Action for No. 3 : Figure 3 shows the magnitude of influence of causal variables in terms of direct and indirect effects, and this provides a clearer picture of such influence. Unlike many previous researches of indicating the magnitude and direction of the relationship in a single direction only. Therefore, this research is deemed to benefit in the national policy planning.

Comment for No. 4: Minor stylistic errors, for example: line 433 begins with), in the formula 13 a Chinese character appeared, 423-427 a few unnecessary spaces

Action for No. 4 : I was done

Comment for No. 5: To check the correctness of the model, I would like to see its verification for previous years.

Action for No. 5 : The following table shows the CO2 emission (Mt CO2 Eq) in the past during 1990 to 2017 in the sector of transportation of Thailand. This information or data can be disclosed to the reviewer as requested. While other information relating to a national security and a national strategy formulation of Thailand, the researcher is not allowed to disclose to any party, and the researcher humbly offers an apology upon such inconvenience caused. Personally, the researcher is responsible in defining national policies and sustainable development plans of Thailand. Thus, it is important the researcher to secure such information for a national security purpose. The researcher would like to ask the reviewers a kind understanding and support to have this study published.

year

CO2 emission (Mt CO2 Eq)

1990

31.89

1991

32.05

1992

32.13

1993

32.55

1994

35.11

1995

36.45

1996

37.12

1997

37.65

1998

38.08

1999

38.62

2000

39.14

2001

40.05

2002

41.23

2003

45.55

2004

46.19

2005

47.52

2006

47.99

2007

48.91

2008

50.57

2009

51.64

2010

58.44

2011

62.17

2012

69.64

2013

72.41

2014

72.69

2015

73.01

2016

73.56

2017

73.94

Comment for No. 6: No conclusion drawn.

Action for No. 6 :  I have made a correction according to your comment, and it can be found in line 554-580.

Comment for No. 7: No applications.

Action for No. 7 :  The researcher has illustrated in Figure 3 line 424 and Table 3 line  439, and drawn a first conclusion as shown in line 555

Best regards

Round  2

Reviewer 3 Report

Most of my comments have been taken into account. Please broaden the literature review including the journal in which the authors want to publish the article. Without this, it is impossible to determine if the manuscript is suitable for this journal.

Author Response

Dear reviewer and Editor

Comment for No.1: Most of my comments have been taken into account. Please broaden the literature review including the journal in which the authors want to publish the article. Without this, it is impossible to determine if the manuscript is suitable for this journal.

Active for No.1 I was done, in line 81-84

Best regards

Professor Dr. Pruethsan Sutthichaimethee
